# Preference about Laws for the Legal Recognition of Same-Sex Relationships in Taiwanese People Before and After Same-Sex Marriage Referenda: A Facebook Survey Study

**DOI:** 10.3390/ijerph17062000

**Published:** 2020-03-18

**Authors:** Cheng-Fang Yen, Nai-Ying Ko, Yu-Te Huang, Mu-Hong Chen, I-Hsuan Lin, Wei-Hsin Lu

**Affiliations:** 1Department of Psychiatry, School of Medicine, College of Medicine, Kaohsiung Medical University, Kaohsiung 80708, Taiwan; chfaye@cc.kmu.edu.tw; 2Department of Psychiatry, Kaohsiung Medical University Hospital, Kaohsiung 80708, Taiwan; 3Departments of Nursing, College of Medicine, National Cheng Kung University and Hospital, Tainan 70101, Taiwan; nyko@mail.ncku.edu.tw; 4Center of Infection Control, National Cheng Kung University Hospital, Tainan 70101, Taiwan; 5Department of Social Work and Social Administration, the University of Hong Kong, Hong Kong RM543, Hong Kong; Yuhuang@hku.hk; 6Department of Psychiatry, Taipei Veterans General Hospital, Taipei 11217, Taiwan; kremer7119@gmail.com; 7Division of Psychiatry, School of Medicine, National Yang-Ming University, Taipei 11221, Taiwan; 8Department of Psychiatry, Yuan’s General Hospital, Kaohsiung 80249, Taiwan; 9Department of Health Business Administration, Meiho University, Pingtung 91202, Taiwan; 10Department of Psychiatry, Ditmanson Medical Foundation Chia-Yi Christian Hospital, Chia-Yi City 60002, Taiwan; 11Department of Senior Citizen Service Management, Chia Nan University of Pharmacy and Science, Tainan 71710, Taiwan

**Keywords:** homosexuality, same-sex marriage, sexual orientation

## Abstract

This study examined the factors related to the preference about laws to legalize same-sex relationships in participants of the first wave of a survey (Wave 1, 23 months before the same-sex marriage referendum) and the second wave of a survey (Wave 2, 1 week after the same-sex marriage referendum) in Taiwan. The data of 3286 participants in Wave 1 and 1370 participants in Wave 2 recruited through a Facebook advertisement were analyzed. Each participant completed an online questionnaire assessing their attitude toward the legal recognition of same-sex relationships, preference about laws to legalize same-sex relationships (establishing same-sex couple laws outside the Civil Code vs. changing the Civil Code to include same-sex marriage laws), belief in the importance of legalizing same-sex relationships, and perceived social attitudes toward the legal recognition of same-sex relationships. The results revealed that those who did not support legalizing same-sex relationships were more likely to prefer establishing same-sex couple laws outside the Civil Code than those who supported the legalization. The form of law preferred to legalize same-sex relationships significantly changed between Wave 1 and Wave 2. Multiple factors, including gender, age, sexual orientation, belief in the importance of legalizing same-sex relationships to human rights and the social status of sexual minorities, and perceived peers’ and families’ attitudes toward the legal recognition of same-sex relationships, were significantly associated with the preference of laws, although these associations varied among heterosexual and non-heterosexual participants and at various stages of the survey.

## 1. Introduction

### 1.1. Same-Sex Marriage Bans and Legal Recognition of Same-Sex Relationships

As a form of structural-level discrimination, same-sex marriage bans not only socially excluded gay and lesbian individuals by differentially targeting them from heterosexual individuals but also deny them the legal, financial, health-related, and other rights associated with marriage [1,2,3]. Research has demonstrated that same-sex marriage bans were associated with increased rates of psychiatric disorders in lesbian, gay, and bisexual (LGB) populations [4] and suicidal behaviors in men who have sex with men (MSM) [5].

The legal recognition of same-sex relationships has been one of the major achievements of human right campaigns in the past three decades. The social and legal recognition of same-sex relationships can reduce discrimination against LGB individuals [6]. A recent study reported that same-sex marriage legalization accelerated the reduction of both implicit and explicit antigay bias in the United States [7]. Same-sex marriage bestows substantial psychological, social, and health benefits to individuals from sexual minorities [8]. Its legalization also significantly reduced the use of and expenditures on mental health care services among MSM [9]. The results of previous studies have supported the positive effect of same-sex marriage legalization on the health status of LGB individuals.

### 1.2. Forms of Laws for the Legal Recognition of Same-Sex Relationships

Since 1989, several countries and regions in the world have enacted various forms of laws such as partnership registration, civil union, and civil partnership to legalize partial rights for same-sex couples. Over the first decade of the 21st century, a total of 10 countries legalized same-sex marriage to bestow same-sex couples the same marriage-related rights as those for heterosexual couples [10]. Of these 10 countries, some, such as The Netherlands, Spain, Norway, Sweden, and Iceland first legalized same-sex relationships under civil union or registered partnership terms before upgrading to same-sex marriage. By contrast, some countries such as Belgium, Canada, South Africa, Portugal, and Argentina implemented same-sex marriage directly through direct initiatives without applying civil unions or registered partnerships first [10]. Currently, the legalization of same-sex marriage through direct initiatives or referendums without the prior legalization of partial rights for same-sex couples has become the mainstream process worldwide, and it bestows greater benefit than civil unions or domestic partnerships [8].

### 1.3. Battle for the Legal Recognition of Same-Sex Relationships in Taiwan

People in Taiwan traditionally regard homosexuality as a challenge to the family obligations mandated in Confucianism, and in particular, they require their offspring to continue the family bloodline. Campaigners for sexual minority rights in Taiwan have strived for the legal recognition of same-sex relationships since the 1980s. The Article 972 of Taiwan’s Civil Code poses a problem for same-sex marriage campaigners by stipulating that “An agreement to marry shall be reached between a male and a female party of their own accord.” Sexual minority right campaigners have previously sued for the recognition of same-sex marriage, but the Court turned down the petition on the grounds that “homosexuality corrupts social values.” A legislator also proposed a marriage equality amendment to the Civil Code in 2013, but in vain [11].

In the past two decades, overall, an attitude of social tolerance toward homosexuality has become widespread in Taiwan, which is mainly accounted for by improvement in education and liberal values related to gender roles [12]. The 2012 Taiwan Social Change Survey showed that for the first time, supporters of same-sex marriage outnumber those who oppose it [13]. In October 2016, a group of legislators proposed again a Marriage Equality Bill, which passed its first court reading and was subsequently considered by the Judiciary and Organic Laws and Statutes Committee. However, because of a lack of support from the ruling party and the main opposition party, the Marriage Equality Bill was not included into the party negotiations for further inspection.

In addition to the debates on whether same-sex relationships should be legalized, what kinds of laws Taiwan should legislate for same-sex relationship are also the focus of debates in the public hearings hold by the Legislative Yuan and mass media. Two forms of laws have been commonly discussed. The supporters of establishing same-sex couple laws outside the Civil Code argued that it takes the rights of same-sex couples into consideration and keeps the rights of heterosexual couples intact, whereas the supporters of changing the Civil Code to include same-sex marriage laws argued that establishing same-sex couple laws outside the Civil Code without changing the Civil Code itself was virtually discrimination against same-sex couples, as were the separate buses for white and black people in operation across the South of the United States in the 1950s [14].

In May 2017, Taiwan’s Council of Grand Justices announced that the current Civil Code that barred same-sex relationship was a violation of human rights to equality and was unconstitutional. It also stipulated that same-sex relationship should be legalized within two years in Taiwan. This announcement brought a new ray of hope to same-sex marriage campaigners. It was reasonably assumed that the Taiwanese government would be adjusting the Civil Code to include same-sex marriage laws for the sake of human right equality. However, the group against same-sex marriage, mainly supported by Christians, looked unfavorably upon the progress of same-sex relationship legalization. In response to the decision of the Council of Grand Justices, the group against same-sex marriage drafted two referendums arguing that legal reforms should be conducted outside the Civil Code without changes being made to the Civil Code itself. These two referendums were the following: “Do you agree that marriage as defined in the Civil Code should be restricted to unions between a man and a woman?” (Case No. 10) and “Do you agree that the protection of the rights of same-sex couples cohabiting on a permanent basis should be conducted through ways other than changes to the Civil Code?” (Case No. 12). By contrast, the group lobbying in favor of marriage equality drafted a referendum (“Do you agree to the protection of same-sex marital rights through marriage as defined in the Civil Code?”—Case No. 14) arguing that separate legislations amount to a form of discrimination. The results of the vote released on November 24, 2018, indicated that 70.12%, 57.60%, and 30.27% of voters supported Case No. 10, Case No. 12, and Case No. 14, respectively, suggesting that the two referendums against same-sex marriage received overwhelming support compared with the referendum supporting same-sex marriage. Finally, the Taiwanese government enacted the Act for Implementation of Judicial Yuan Interpretation No. 748 outside the Civil Code in May 2019. This law was the effort of Taiwanese government to seek a compromise between the Constitutional Court’s interpretation and the referendum results by guaranteeing most of the same rights entailed in a heterosexual marriage for same-sex couples.

Taiwanese people’s preferences of laws shown in the referendums deeply influenced the final result of legislation. It is important to survey what factors related to the preferences of laws and what changes of the preferences happened during the social debates on legalizing same-sex relationship. The results of the survey may provide an explanation for the people’s attitudes toward the legal recognition of same-sex relationships, as well as may provide knowledge and experiences for other countries that may hold referendums to determine what forms of laws should be established for the legal recognition of same-sex relationships in future.

### 1.4. Aims and Hypotheses of This Study

The present study used data from the Investigation on the Attitude Toward Same-Sex Marriage in Taiwan, which is a two-wave online survey of Taiwanese people’s attitudes toward same-sex marriage [15,16]. The first wave (Wave 1) was conducted from January 1 to 31, 2017, 1 week after the first reading of the Marriage Equality Bill was passed in the Legislative Yuan and 23 months before the referenda for the legalization of same-sex relationships. The second wave (Wave 2) was conducted from December 1 to 31, 2018, 1 week after the referenda.

The three aims and corresponding hypotheses of the present study are described below. The first aim was to compare the rates of preference about laws legalizing same-sex relationships between people who supported the legalization of same-sex relationships in Taiwan and those who did not. Given that establishing same-sex couple laws outside the Civil Code is virtually a discrimination against same-sex couples, we hypothesized that those who did not support the legalization of same-sex relationships preferred establishing same-sex couple laws outside the Civil Code and reserving the rules on marriage as stipulated in the Civil Code for heterosexual people only.

The second aim was to compare the rates of preference of laws legalizing same-sex relationships among people who supported the legalization of same-sex relationships between the first and second waves of the survey. Because of the results of the referendums and the social hostility toward sexual minorities provoked by the rumors spread by the anti-gay group supported by Christians, we hypothesized that the rate of participants that preferred changing the Civil Code to include same-sex marriage laws significantly dropped between the first and second waves of the survey.

The third aim was to examine the factors related to the preference of laws legalizing same-sex relationships among heterosexual and non-heterosexual people who supported the legalization of same-sex relationships in the first and second waves of the survey. Because that legal recognition of same-sex relationship is directly related to the rights of non-heterosexual people, it is apparent that the aspects and attitudes toward this issue in non-heterosexual people will be different from those in heterosexual people; therefore, the present study examined the factors related to the preference of laws in heterosexual and non-heterosexual people separately. Given that individuals’ attitudes may be influenced by their environment [17], we hypothesized that differences existed in the demographic factors, personal belief in the importance of legalizing same-sex relationships, and the perceived attitudes of others toward the legalization of sex-same relationships between participants, leading to their preference of different legal recourse to legalize same-sex relationships.

## 2. Methods

### 2.1. Participants

The method used to recruit participants in the present study was described elsewhere [16]. In brief, participants aged at least 20 years were recruited to participate in the two-wave online survey through a Facebook advertisement. The Facebook advertisement included a headline, main text, pop-up banner, and weblink to the study questionnaire website. The advertisement appeared in the news feed on Facebook, which is a streaming list of updates from the user’s connections and advertisers. News feed advertisements have proven more effective in terms of recruitment metrics for studies [18]. We targeted Facebook users by location (Taiwan) and language (Chinese). The deduplication protocol used in the present study to identify multiple submissions and preserve data integrity included the cross-validation of the eligibility of key variables, examination of discrepancies in key data, and screening of unusually fast completion times <10 min [19]. Moreover, each Internet protocol address could be used only once to complete the online questionnaire.

Participants were not given any incentives for participation. This study was approved by the Institutional Review Board (IRB) of Kaohsiung Medical University Hospital. The study design involved the anonymity of respondents’ online response to the recruitment advertisement and questionnaire, which enabled the respondents to freely decide whether to join or not and ensured their personal information was kept secure. Owing to the anonymity of participants, we could not determine precisely how many participants responded to both surveys. Therefore, the data of the two waves of the survey were analyzed independently. The IRB thus agreed that this study did not require an informed consent to be filled by the respondents. In total, 3286 participants in Wave 1 and 1370 participants in Wave 2 were recruited.

### 2.2. Measures

#### 2.2.1. Preference of Laws for the Legal Recognition of Same-Sex Relationships

We asked the participants the following questions: “Which forms of law do you prefer for the legal recognition of same-sex relationships in Taiwan?” The two options were “establishing same-sex couple laws outside the Civil Code and not change the Civil Code itself” and “changing the Civil Code to include same-sex marriage laws.” The participants were classified into two groups according to their response to the questions for further comparison.

#### 2.2.2. In Favor of the Legal Recognition of Same-Sex Relationships

The question “To what degree do you support the legal recognition of same-sex relationships?” was used to evaluate participants’ attitude toward the legal recognition of same-sex relationships in the week preceding this survey. Participants indicated their level of support on a 5-point Likert scale ranging from 0 (*very low*) to 4 (*very high*). Participants who scored 0 to 2 points were classified into the group against and those who scored 3 or 4 points were classified into the group in favor of the legal recognition of same-sex relationships.

#### 2.2.3. Personal Belief toward the Importance of the Legal Recognition of Same-Sex Relationships

We used two questions to evaluate the importance attached to the legal recognition of same-sex relationships: “To what degree do you believe the legal recognition of same-sex relationships is important to human right equality?” and “To what degree do you believe the legal recognition of same-sex relationships is important to the social status of sexual minorities?”. Participants rated the level of importance of legalizing same-sex relationships on a 5-point Likert scale ranging from 0 (*very low*) to 4 (*very high*). For the purpose of this study, participants who scored 0 to 3 points and those who scored 4 points were respectively classified into the group who did not deem legalization of same-sex relationships important and that who deemed legalization of same-sex relationships important for human right equality and the social status of sexual minorities.

#### 2.2.4. Perceived Social Attitudes toward the Legal Recognition of Same-Sex Relationships

We used two questions to evaluate how participants perceived their heterosexual friends’ and family members’ attitudes toward the legalization of same-sex relationships (“To what degree do your heterosexual friends support the legalization of same-sex relationships?” and “To what degree do your family members support the legalization of same-sex relationships?”) in the preceding week. Participants indicated the level of perceived social attitudes in favor of the legalization of same-sex relationships on a 5-point Likert scale ranging from 0 (*very low*) to 4 (*very high*). For the purpose of this study, participants who scored 0 to 2 points were classified into the group that perceived as unfavorable the attitudes of their heterosexual friends and family members toward the legalization of same-sex relationships and those who scored 3 or 4 points were classified into the group that perceived favorable attitudes toward the legalization of same-sex relationships from their heterosexual friends and family members.

#### 2.2.5. Demographic Variables

Data on the participants’ gender (male, female, and transgender), age (20–29 years, 30–39 years, and 40 years or older), and sexual orientation (heterosexual, bisexual, homosexual, pansexual, asexual, and questioning) were collected. According to their sexual orientation, participants were classified into the non-heterosexual (including bisexual, homosexual, and others) or heterosexual group.

### 2.3. Statistical Analysis

Data analysis was performed using SPSS Version 20.0 (SPSS Inc., Chicago, IL, USA). The rates of various laws for legalizing same-sex relationships (establishing same-sex couple laws outside the Civil Code and not changing the Civil Code itself vs. changing the Civil Code so that it includes same-sex marriage laws) were compared between participants who did not support and those who supported the legalization of same-sex relationships in Wave 1 and Wave 2 surveys using chi-square test. Moreover, preference of laws for legalizing same-sex relationships among participants supporting the legalization of same-sex relationships were compared between Wave 1 and Wave 2 surveys using chi-square test. A *p* value of <0.05 was considered statistically significant. Furthermore, gender, age, personal belief toward the importance of the legal recognition of same-sex relationships to human right equality and the social status of sexual minorities, and perceived social attitudes in favor of the legalization of same-sex relationships were compared between participants with various preferences of laws in Wave 1 and Wave 2 surveys using chi-square test. Because of the existence of multiple comparisons, *p* values of <0.008 (0.05/6) and <0.007 (0.05/7) were considered statistically significant for heterosexual and non-heterosexual participants, respectively. The significant factors were further selected into multivariate logistic regression analysis to examine their relationships with preference of laws for legalizing same-sex relationship.

## 3. Results

### 3.1. Difference in Preference of Laws between Participants Supporting the Legal Recognition of Same-Sex Relationships at Various Degrees

In total, 251 participants in Wave 1 and 111 participants in Wave 2 reported that they did not support the legal recognition of same-sex relationships. Moreover, 3035 participants in Wave 1 and 1259 participants in Wave 2 reported that they supported the legal recognition of same-sex relationships. Table 1 presents the results of the comparison of preferences of laws for the legal recognition of same-sex relationships between participants who did not support and who those who supported the legalization of same-sex relationships. The results indicated that in both Wave 1 and Wave 2 of the survey, those who did not support the legalization of same-sex relationships were more likely to prefer establishing same-sex couple laws outside the Civil Code than those who supported the legalization of same-sex relationships (*p* < 0.001).

### 3.2. Preference of Laws for the Legal Recognition of Same-Sex Relationships: Changes between Wave 1 and Wave 2

Table 2 lists the demographic data and preference of laws for the legal recognition of same-sex relationships of participants who supported the legalization of same-sex relationships between Wave 1 and Wave 2. 

The rate of heterosexual participants who preferred changing the Civil Code to include same-sex marriage laws dropped from 64.5% in Wave 1 to 51.5% in Wave 2, whereas the rate of heterosexual participants who preferred keeping the Civil Code unchanged and establishing same-sex couple laws outside the Civil Code increased from 35.5% in Wave 1 to 48.5% in Wave 2 (*p* < 0.001). The rate of non-heterosexual participants who preferred changing the Civil Code to include same-sex marriage laws dropped from 78.9% in Wave 1 to 58.4% in Wave 2, whereas the rate of non-heterosexual participants who preferred keeping the Civil Code unchanged and establishing same-sex couple laws outside the Civil Code increased from 21.1% in Wave 1 to 41.6% in Wave 2 (*p* < 0.001).

### 3.3. Factors Related to Preferences of Laws for the Legal Recognition of Same-Sex Relationships

#### 3.3.1. Heterosexual Participants

Table 3 presents the gender, age, personal belief toward the importance of same-sex marriage, and perceived social attitudes toward the legalization of same-sex relationships of heterosexual participants who supported the legalization of same-sex relationships with various preferences of laws for the legalization of same-sex relationships in Wave 1 and Wave 2. The results indicated that in Wave 1, heterosexual males were more likely to prefer changing the Civil Code to include same-sex marriage laws than heterosexual females. In Wave 2, heterosexual participants aged 20–29 years were more likely to prefer changing the Civil Code to include same-sex marriage laws than those aged 40 or older. No age and gender differences were observed in the preference of laws among heterosexual participants in Wave 1 and Wave 2, respectively.

Heterosexual participants who rated legalizing same-sex relationship as important to human rights equality and social status of sexual minorities were more likely to prefer changing the Civil Code to include same-sex marriage laws instead of establishing same-sex couple laws outside the Civil Code than those who did not rate legalizing same-sex relationships as important in both Wave 1 and Wave 2.

Heterosexual participants who perceived peers’ and families’ attitudes toward the legal recognition of same-sex relationships as favorable were more likely to prefer changing the Civil Code to include same-sex marriage laws instead of establishing same-sex couple laws outside the Civil Code than those who perceived peers’ and families’ attitudes as unfavorable in Wave 1 of the survey. However, no difference in perceived peers’ and families’ attitudes toward the legal recognition of same-sex relationships was observed between heterosexual participants preferring various forms of laws in Wave 2 of the survey.

The significant factors were further selected into logistic regression analysis (Table 4). The results indicated that in the Wave 1 survey, heterosexual participants who believed in the importance of legal recognition of same-sex relationships to human right equality and social status of sexual minority were more likely to prefer changing the Civil Code to include same-sex marriage laws instead of establishing same-sex couple laws outside the Civil Code, whereas heterosexual participants who perceived families’ unfavorable attitude toward legal recognition of same-sex relationship were more likely to prefer establishing same-sex couple laws outside the Civil Code instead of changing the Civil Code to include same-sex marriage laws. Their interactions, including importance to human right equality x importance to social status of sexual minority (*p* = 0.113), importance to human right equality x perceived families’ unfavorable attitude (*p* = 0.787), and importance to social status of sexual minority x perceived families’ unfavorable attitude (*p* = 0.455) were not significantly associated with preference of laws in multivariate logistic regression analysis, indicating that the interaction effects among these three factors were not significant. Compared with the heterosexual participants with the age of 40 or older in the Wave 2 survey, the heterosexual participants with the age of 30–39 were more likely to prefer establishing same-sex couple laws outside the Civil Code in the Wave 2 survey.

#### 3.3.2. Non-Heterosexual Participants

Table 5 presents the gender, age, sexual orientation, personal belief toward the importance of same-sex marriage, and perceived social attitudes toward the legalization of same-sex relationships of non-heterosexual participants who supported the legalization of same-sex relationships with various preference of laws for the legal recognition of same-sex relationships in Wave 1 and Wave 2. The results indicated that in Wave 1, non-heterosexual participants who were male and homosexual were more likely to prefer changing the Civil Code to include same-sex marriage laws than those who were female and bisexual or of other sexual orientation. No age difference was observed in the preference of laws among non-heterosexual participants in Wave 1. No difference in gender, age, or sexual orientation was observed in the preference of laws among non-heterosexual participants in Wave 2.

Non-heterosexual participants who rated the legal recognition of same-sex relationships as important to human rights equality and social status of sexual minorities were more likely to prefer changing the Civil Code to include same-sex marriage laws instead of establishing same-sex couple laws outside the Civil Code than those who did not rate the legal recognition of same-sex relationships as important in both Wave 1 and Wave 2 of the survey. No difference in the perceived peers’ and families’ attitudes toward the legal recognition of same-sex relationships was observed in the preference of laws among non-heterosexual participants in Wave 1 and 2.

The significant factors were further selected into logistic regression analysis (Table 6). The results indicated that in both Wave 1 and Wave 2 surveys, non-heterosexual participants who believed in the importance of legal recognition of same-sex relationships to human right equality were more likely to prefer changing the Civil Code to include same-sex marriage laws instead of establishing same-sex couple laws outside the Civil Code.

## 4. Discussion

The present study revealed that those who did not support the legalization of same-sex relationships were more likely to prefer establishing same-sex couple laws outside the Civil Code than those who supported the legalization of same-sex relationships. The rates of both heterosexual and non-heterosexual participants who preferred changing the Civil Code to include same-sex marriage laws significantly dropped from Wave 1 to Wave 2 of the survey, whereas the rates of heterosexual and non-heterosexual participants who preferred keeping the Civil Code unchanged and establishing same-sex couple laws outside the Civil Code significantly increased. Moreover, multiple factors including age, beliefs in the importance of legalizing same-sex relationships to human right and the social status of sexual minorities, and perceived families’ attitudes toward the legal recognition of same-sex relationships were significantly associated with the preference of laws for legalizing same-sex relationships, although these associations varied among heterosexual and non-heterosexual participants and according to the various waves of the survey.

### 4.1. People against the Legal Recognition of Same-Sex Relationships Favored Establishing Same-Sex Couple Laws outside the Civil Code

The claim about establishing same-sex couple laws outside the Civil Code and keeping the Civil Code unchanged for heterosexual individuals earned overwhelming support in the referendums held on November 24, 2018 in Taiwan. The advocates of this idea preached that establishing same-sex couple laws outside the Civil Code had numerous advantages, including “preserving traditional family values,” “reducing the shock of legalizing same-sex relationships on the Taiwanese society,” and “simplifying the steps of legislation.” They also took Germany as example and insisted that establishing same-sex couple laws outside the Civil Code was the best way to initiate the legal recognition of same-sex relationships. However, the advocates of establishing same-sex couple laws outside the Civil Code deliberately omitted the fact that changing the Civil Code to include same-sex marriage is the mainstream adopted procedure worldwide currently and that it bestows greater benefit than civil unions or domestic partnerships [8]. The present study revealed that unfavorable attitude toward the legalization of same-sex relationships was significantly associated with the preference of establishing same-sex couple laws outside the Civil Code, indicating the possibility that the motive for resisting the legal recognition of same-sex relationships and equality may have consciously or unconsciously brought about the preference of establishing same-sex couple laws outside the Civil Code.

According to Social Exchange Theory [20,21], the concepts of equity and distributive justice may partially account for the result. All social systems evolve mechanisms for distributing valued resources and for allocating rights, responsibilities, costs, and burdens [20]. Theories of distributive justice specify the conditions under which particular distributions are perceived to be “just” or “fair” [20]. Compared with heterosexual relationships, same-sex relationships may have many advantages and fewer costs or risks. For example, gay men cannot become pregnant, and hence it is unnecessary for them to worry over forced marriages, unwanted abortions, child support requirements, and the formation of relationships based on having a mutual child rather than mutual love. In the debates on legalizing same-sex relationships, heterosexual individuals may be concerned with the possible rewards and costs; the results of weighing may influence the degree of supporting legalizing same-sex relationship and preference of laws [22].

### 4.2. Change of Preferences of Laws for the Legal Recognition of Same-Sex Relationships before and after the Referendums

The present study revealed that although most participants who supported the legal recognition of same-sex relationships preferred changing the Civil Code to include same-sex marriage laws, the rate dropped significantly from Wave 1 to Wave 2 of the survey. By contrast, the rate of individuals that preferred establishing same-sex couple laws outside the Civil Code increased significantly, especially in non-heterosexual participants (from 21.1% to 41.6%). In the referendums held on November 24, 2018, 70.12% of voters supported the establishment of same-sex couple laws outside the Civil Code, but only 30.27% of voters supported changing the Civil Code to include same-sex marriage. The results of the referendums deeply discouraged those in favor of the legal recognition of same-sex relationships and shook their preference of laws to legalize same-sex relationships. Non-heterosexual individuals may have felt particularly attacked by the results of the referendums and social hostility provoked by the rumors spread by the anti-gay group.

### 4.3. Beliefs in the Importance of Legalizing Same-Sex Relationships and Preference of Laws

The present study revealed that the belief in the importance of legalizing same-sex relationships to human rights and social status of sexual minorities had a significant role in the preference of changing the Civil Code to include same-sex marriage laws. The importance of legalizing same-sex relationships to human rights and social status of sexual minorities has been supported by the results of previous studies [6,7]. For heterosexual individuals who are friendly to sexual minorities, the legal recognition of same-sex relationships is also a symbol of social justice. Therefore, the preference of changing the Civil Code to include same-sex marriage laws instead of establishing same-sex couple laws outside the Civil Code is an act of support to the concept that all humans are equal before the law. However, the present study revealed that the numbers of participants who believed in the importance of legalizing same-sex relationships to human rights but preferred establishing same-sex couple laws outside the Civil Code increased from Wave 1 to Wave 2 (heterosexual: from 28.5% to 44.6%; non-heterosexual: from 18.2% to 39.6%). Moreover, the number of participants who believed in the importance of legalizing same-sex relationships to the social status of sexual minorities but preferred establishing same-sex couple laws outside the Civil Code increased (heterosexual: from 45.7% to 52.7%; non-heterosexual: from 25.4% to 43.5%). Given that the groups against legalization of same-sex relationship claimed that the legalization of same-sex relationships would lead to a widespread outbreak of human immunodeficiency virus infection, depopulation in Taiwan, and the deterioration of traditional family values before the referendums, it is reasonable that these changes of preference of laws may reflect the negative effects of the claims by the anti-gay group to oppose legal recognition of same-sex relationships.

### 4.4. Perceived Social Attitudes and Preference of Laws

The present study revealed that heterosexual participants who perceived families’ attitudes toward the legal recognition of same-sex relationships as favorable were more likely to prefer changing the Civil Code to include same-sex marriage laws than those who perceived families’ attitudes as unfavorable only in Wave 1 but not in Wave 2. The legal recognition of same-sex relationships might be an unfamiliar issue to most heterosexual individuals. Therefore, perceived favorable attitudes from families might support them to stand for a law that views the rights of sexual minorities as those of heterosexual individuals. Compared with perceived attitudes from peers, perceived attitudes from families was a more significant factor of preferring changing the Civil Code to include same-sex marriage laws, indicating that families’ attitudes may have a deep and far influence on heterosexual individuals regarding human rights of sexual minority people.

However, the association between perceived social attitudes and preference of laws in heterosexual individuals became nonsignificant in Wave 2 of the survey. It is possible that heterosexual individuals received mass information regarding the legalization of same-sex relationships during the referendums, and the influence of perceived families’ attitudes decreased. The sources of information that had the most influence on heterosexual individuals’ preference of law to legalize same-sex relationships warrant further investigation.

### 4.5. Gender, Age and Preference of Laws

Research has indicated that heterosexual men have more negative attitudes toward homosexuality than do heterosexual women [23,24]. However, the present study focused on a group of participants supporting the legal recognition of same-sex relationships and revealed no gender difference in the preference of laws for legalizing same-sex relationship in multivariate logistic regression. The results indicated that the beliefs in the importance of and perceived families’ attitudes legalization of same-sex relationship have a more significant role for preference of laws than gender. The present study indicated the heterosexual participants with the age of 30–39 were more likely to prefer establishing same-sex couple laws outside the Civil Code than those with the age of 40 or older in the Wave 2 survey. No difference of sexual orientation in preference of laws among non-heterosexual participants. Further study is warranted to replicate the results found in this study.

### 4.6. Limitations

The present study has some limitations. First, although recruiting participants through Facebook can ensure a large number of participants quickly, cheaply, and with minimal effort compared with mail and phone recruitment, access to Facebook is not yet universal, and all people are not equally motivated to use Facebook [25]. Second, the cross-sectional study design limited the possibility to determine the causal relationships between perceived social attitude toward the legal recognition of same-sex relationships and the preference of laws. Third, we were unable to determine the mechanisms underlying the changes in the preference of laws to legalize same-sex relationships before and after the same-sex marriage referendums. Fourth, the distributions of heterosexual and non-heterosexual participants in this study were not in accordance with real conditions. Moreover, like other studies recruiting participants via Facebook [26,27], the participants in this study tended to be young, and only 12.9% and 19.6% of the participants were 40 or older in the Wave 1 and Wave 2 surveys, respectively. Age has been identified as a factor significantly related to the level of tolerance for homosexuality in Taiwan [12].

## 5. Conclusions

Taiwan is the first Asian country to have formally considered the legal recognition of same-sex relationships, although the law was established outside the Civil Code. The present study discovered several factors related to the preference of laws to legalize same-sex relationships, including attitudes toward the legal recognition of same-sex relationships, beliefs in the importance of legalizing same-sex relationships to human rights and the social status of sexual minorities, perceived peers’ and families’ attitudes toward the legal recognition of same-sex relationships, gender, age, and sexual orientation. In future, other countries might hold referendums to determine whether same-sex relationships should be legalized or what forms of laws should be established for the legal recognition of same-sex relationships. Groups lobbying in favor of marriage equality may take the factors highlighted in the present study into consideration and develop programs accordingly to enhance people’s understanding of the importance of laws legalizing same-sex relationships. Only comprehensive understanding can lead to the establishment of laws providing equality to sexual minorities. There are still many issues of legalizing same-sex relationships, for example, relationship stability in same-sex couples rearing children and the development of children raised by same-sex couples warranted further study. For example, some studies found that psychological adjustment among children was not different on the basis of parental sexual orientation [28,29], whereas other studies found that the rates of illegal drug use [30] and depression [31] are higher among children of same-sex parents than among children of comparable heterosexual parents. Especially, cultural and social differences may exist in these issues of legalizing same-sex relationships. Further study in Taiwan may provide empirical experiences to the field of study on sexual minority.

## Figures and Tables

**Table 1 ijerph-17-02000-t001:** Differences in the preference of laws for the legal recognition of same-sex relationships between participants with various attitudes toward the legalization of same-sex relationships.

Wave/Support	*n*	Same-Sex Couple Law outside the Civil Code	Same-Sex Marriage Law in the Civil Code	Χ^2^	*p*
Wave 1	3286				
Supporting legalization of same-sex relationships, *n* (%)					
Yes	3035	818 (27.0)	2217 (73.0)	**246.891**	**<0.001**
No	251	187 (74.5)	64 (25.5)		
Wave 2	1370				
Supporting legalization of same-sex relationships, *n* (%)					
Yes	1259	555 (44.1)	704 (55.9)	**30.107**	**<0.001**
No	111	79 (71.2)	32 (28.8)		

**Table 2 ijerph-17-02000-t002:** Comparison of preferences of laws for the legal recognition of same-sex relationships in heterosexual and non-heterosexual participants supporting the legal recognition of same-sex relationships between Wave 1 and Wave 2.

Wave	Heterosexual					Non-Heterosexual				
	*n*	Same-Sex Couple Law outside the Civil Code	Same-Sex Marriage Law in the Civil Code	χ^2^	*p*	*n*	Same-Sex Couple Law outside the Civil Code	Same-Sex Marriage Law in the Civil Code	χ^2^	*p*
Wave 1 survey	1234	438 (35.5)	796 (64.5)	**23.547**	**<0.001**	1801	380 (21.1)	1421 (78.9)	**117.217**	**<0.001**
Wave 2 survey	456	221 (48.5)	235 (51.5)			803	334 (41.6)	469 (58.4)		

**Table 3 ijerph-17-02000-t003:** Comparisons of gender, age, personal belief, and perceived social attitudes between heterosexual participants supporting the legalization of same-sex relationships with various preference of forms of laws in Wave 1 and Wave 2.

Variables	Heterosexual									
	Wave 1					Wave 2				
	*N* = 1234	Same-Sex Couple Law outside the Civil Code	Same-Sex Marriage Law in the Civil Code	χ^2^	*p*	*N* = 456	Same-Sex Couple Law outside the Civil Code	Same-Sex Marriage Law in the Civil Code	χ^2^	*p*
Gender, *n* (%)										
Male	250	62 (24.8)	188 (75.2)	**16.979**	**<0.001**	89	37 (41.6)	52 (58.4)	2.103	0.147
Female	971	373 (38.4)	598 (61.6)			367	184 (50.1)	183 (49.9)		
Transgender	13	3 (23.1)	10 (76.9)			0	0	0		
Age, *n* (%)										
40 years or older	204	89 (43.6)	115 (56.4)	7.204	0.027	145	86 (59.3)	59 (40.7)	**13.391**	**0.001**
30–39 years	446	154 (34.5)	292 (65.5)			168	81 (48.2)	87 (51.8)		
20–29 years	584	195 (33.4)	389 (66.6)			143	54 (37.8)	89 (62.2)		
Legal recognition of same-sex relationships to human right equality, *n* (%)										
No important	191	141 (73.8)	50 (26.2)	**144.988**	**<0.001**	73	50 (68.5)	23 (31.5)	**13.959**	**<0.001**
Important	1043	297 (28.5)	746 (71.5)			383	171 (44.6)	212 (55.4)		
Legal recognition of same-sex relationships to social status of sexual minority, *n* (%)										
No important	296	169 (57.1)	127 (42.9)	**79.353**	**<0.001**	111	68 (61.3)	43 (38.7)	**9.619**	**0.002**
Important	938	269 (28.7)	669 (71.3)			345	153 (44.3)	192 (55.7)		
Perceived peers’ attitude toward legal recognition of same-sex relationships, *n* (%)										
Favorable	976	320 (32.8)	656 (67.2)	**14.945**	**<0.001**	310	144 (46.5)	166 (53.5)	1.571	0.210
Unfavorable	258	118 (45.7)	140 (54.3)			146	77 (52.7)	69 (47.3)		
Perceived families’ attitude toward legal recognition of same-sex relationships, *n* (%)										
Favorable	531	155 (29.2)	376 (70.8)	**16.179**	**<0.001**	167	82 (49.1)	85 (50.9)	0.043	0.836
Unfavorable	703	283 (40.3)	420 (59.7)			289	139 (48.1)	150 (51.9)		

**Table 4 ijerph-17-02000-t004:** Factors related to preference of establishing same-sex marriage law in the Civil Code among heterosexual participants supporting the legalization of same-sex relationships in Wave 1 and Wave 2 ^a^.

	Heterosexual							
	Wave 1				Wave 2			
	Wals χ^2^	*p*	OR	95% CI	Wals χ^2^	*p*	OR	95% CI
Gender ^b^								
Female	0.048	0.826	1.161	0.305–4.420				
Transgender	0.539	0.463	0.613	0.166–2.265				
Age ^c^								
30–39 years					**10.050**	**0.002**	**0.456**	**0.281–0.741**
20–29 years					2.886	0.089	0.672	0.424–1.063
Beliefs in the importance of legal recognition of same-sex relationships to human right equality	**59.824**	**<0.001**	**4.855**	**3.253–7.245**	2.804	0.094	1.844	0.901–3.774
Beliefs in the importance of legal recognition of same-sex relationships to social status of sexual minority	**10.179**	**0.001**	**1.703**	**1.228–2.362**	1.161	0.281	1.384	0.766–2.499
Perceived peers’ unfavorable attitude toward legal recognition of same-sex relationships	2.732	0.098	0.768	0.562–1.050				
Perceived families’ unfavorable attitude toward legal recognition of same-sex relationships	**5.591**	**0.018**	**0.730**	**0.562–0.948**				

^a^: Establishing same-sex couple law outside the Civil Code as the reference; ^b^: Male as the reference; ^c^: 40 years or older as the reference.

**Table 5 ijerph-17-02000-t005:** Comparison of gender, age, sexual orientation, personal belief, and perceived social attitudes between non-heterosexual participants supporting the legalization of same-sex relationships with various preferences of forms of laws in Wave 1 and Wave 2.

Variables	Non-Heterosexual									
	Wave 1					Wave 2				
	*N* = 1801	Same-Sex Couple Law outside the Civil Code	Same-Sex Marriage Law in the Civil Code	χ^2^	*p*	*N* = 803	Same-Sex Couple Law outside the Civil Code	Same-Sex Marriage Law in the Civil Code	χ^2^	*p*
Gender, *n* (%)										
Male	863	151 (17.5)	712 (82.5)	**13.044**	**0.001**	371	157 (42.3)	214 (57.7)	0.267	0.875
Female	906	222 (24.5)	684 (75.5)			403	166 (41.2)	237 (58.8)		
Transgender	32	7 (21.9)	25 (78.1)			29	11 (37.9)	18 (62.1)		
Age, *n* (%)										
40 years or older	142	37 (26.1)	105 (73.9)	5.421	0.061	75	37 (49.3)	38 (50.7)	5.686	0.058
30–39 years	599	138 (23.0)	461 (77.0)			269	122 (45.4)	147 (54.6)		
20–29 years	1060	205 (19.3)	855 (80.7)			459	175 (38.1)	284 (61.9)		
Sexual orientation, *n* (%)										
Homosexual	1154	204 (17.7)	950 (82.3)	**22.698**	**<0.001**	517	203 (39.3)	314 (60.7)	3.833	0.147
Bisexual	414	111 (26.8)	303 (73.2)			190	84 (44.2)	106 (55.8)		
Others ^a^	233	65 (27.9)	168 (72.1)			96	47 (49.0)	49 (51.0)		
Legal recognition of same-sex relationships to human right equality, *n* (%)										
No important	110	72 (65.5)	38 (34.5)	**138.452**	**<0.001**	55	38 (69.1)	17 (30.9)	**18.376**	**<0.001**
Important	1691	308 (18.2)	1383 (81.8)			748	296 (39.6)	452 (60.4)		
Legal recognition of same-sex relationships to social status of sexual minority, *n* (%)										
No important	283	93 (32.9)	190 (67.1)	**27.906**	**<0.001**	91	52 (57.1)	39 (42.9)	**10.214**	**0.001**
Important	1518	287 (18.9)	1231 (81.1)			712	282 (39.6)	430 (60.4)		
Perceived peers’ attitude toward legal recognition of same-sex relationships, *n* (%)										
Favorable	1450	291 (20.1)	1159 (79.9)	4.745	0.029	571	233 (40.8)	338 (59.2)	0.506	0.477
Unfavorable	351	89 (25.4)	262 (74.6)			232	101 (43.5)	131 (56.5)		
Perceived families’ attitude toward legal recognition of same-sex relationships, *n* (%)										
Favorable	680	138 (20.3)	542 (79.7)	0.426	0.514	263	113 (43.0)	150 (57.0)	0.303	0.582
Unfavorable	1121	242 (21.6)	879 (78.4)			540	221 (40.9)	319 (59.1)		

^a^: Pansexual, asexual, and questioning.

**Table 6 ijerph-17-02000-t006:** Factors related to preference of establishing same-sex marriage law in the Civil Code among non-heterosexual participants supporting the legalization of same-sex relationships in Wave 1 and Wave 2 ^a^.

	Non-Heterosexual							
	Wave 1				Wave 2			
	Wals χ^2^	*p*	OR	95% CI	Wals χ^2^	*p*	OR	95% CI
Gender ^b^								
Female	0.607	0.436	1.413	0.592–3.371				
Transgender	0.005	0.942	1.032	0.439–2.429				
Sexual orientation ^c^								
Bisexual	2.634	0.105	1.355	0.939–1.956				
Others	0.042	0.838	0.961	0.656–1.407				
Beliefs in the importance of legal recognition of same-sex relationships to human right equality	**75.619**	**<0.001**	**8.951**	**5.462–14.670**	**9.240**	**0.002**	**2.801**	**1.442–5.440**
Beliefs in the importance of legal recognition of same-sex relationships to social status of sexual minority	0.300	0.584	0.900	0.618–1.311	1.584	0.208	1.387	0.833–2.307

^a^: Establishing same-sex couple law outside the Civil Code as the reference; ^b^:male as the reference; ^c^: Homosexuality as the reference.

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
