# Peer review of "Preference about Laws for the Legal Recognition of Same-Sex Relationships in Taiwanese People Before and After Same-Sex Marriage Referenda: A Facebook Survey Study"

_ijerph, 2020, doi:10.3390/ijerph17062000_

Round 1
Reviewer 1 Report
This paper deserves to be published because it deals with the important roles that continued perceptions of same-sex couples in derogatory terms have on how people (heterosexuals or non-heterosexuals) are likely to vote when they are asked to participate in referendums deciding whether same-sex marriages must be legalized in or outside of Taiwan's Civil Code. The paper's authors use sound research and methodology to reveal all the contradictions that are apparent in how supporters of change outside the Civil Code approach the issue. Using strong statistics that are cogently explained, the authors make a strong case that both the research and the conclusion support.
In addition to the above, I like the clarity of the language of the essay even if I recommend that the authors proofread the text a few more times just to catch occasional typos and unnecessary words that may lurk in it. A few of these typos and words are highlighted in the attached version of the article.

Author Response
We appreciate your comments on our manuscript. As discussed below, we have revised our manuscript with underlines according to the reviewers. The following responses have been prepared to address your comments in a point-by-point fashion. Please let us know if there is anything else we should provide.
Comment
This paper deserves to be published because it deals with the important roles that continued perceptions of same-sex couples in derogatory terms have on how people (heterosexuals or non-heterosexuals) are likely to vote when they are asked to participate in referendums deciding whether same-sex marriages must be legalized in or outside of Taiwan's Civil Code. The paper's authors use sound research and methodology to reveal all the contradictions that are apparent in how supporters of change outside the Civil Code approach the issue. Using strong statistics that are cogently explained, the authors make a strong case that both the research and the conclusion support.
Response
Thank you for your comment.
Comment
In addition to the above, I like the clarity of the language of the essay even if I recommend that the authors proofread the text a few more times just to catch occasional typos and unnecessary words that may lurk in it. A few of these typos and words are highlighted in the attached version of the article.
Response
We appreciate your careful review. We corrected the typos and deleted unnecessary words that you pointed out. We also thank for your positive comments on our study.
Reviewer 2 Report
The article is interesting but is complicate to see all the data in the table; also is better to have more information related with the backgrund and also to explain why it was neccesary the study.
Author Response
We appreciate your comments on our manuscript. As discussed below, we have revised our manuscript with underlines according to the reviewers. The following responses have been prepared to address your comments in a point-by-point fashion. Please let us know if there is anything else we should provide.
Comment
The article is interesting but is complicate to see all the data in the tables.
Response
Thank you for your comments. We revised Table 3 and Table 4 in the revised manuscript, and their contents can be seen now.
Comment
Also is better to have more information related with the background and also to explain why it was necessary the study.
Response
- Thank you for your comments. We added more information regarding traditional opinions on homosexuality among people in Taiwan and the change of social attitudes toward homosexuality and same-sex relationship in the past two decades as below. Please refer to line 80-82 and line 89-92.
“People in Taiwan traditionally regard homosexuality as a challenge to the family obligations mandated in Confucianism, and in particular, they require their offspring to continue the family bloodline.”
”In the past two decades, overall, an attitude of social tolerance toward homosexuality has become widespread in Taiwan, which is mainly accounted for by improvement in education and liberal values related to gender roles [12]. The 2012 Taiwan Social Change Survey showed that for the first time, supporters of same-sex marriage outnumber those who oppose it [13].”
- We added an paragraph in Introduction section to introduce why the present study was necessary. Please refer to line 126-133.
“Social debates on legal recognition of same-sex relationship in Taiwan lasted for at two years. For most of people in Taiwan, it is the first time to seriously consider legal recognition of same-sex relationship. It is a good opportunity to examine what factors might influence Taiwanese people’s preference of laws legalizing same-sex relationships. The results of the survey may provide an explanation for the people’s attitudes toward the legal recognition of same-sex relationships, as well as may provide knowledge and experiences for other countries that may hold referendums to determine what forms of laws should be established for the legal recognition of same-sex relationships in future.”
Reviewer 3 Report
While public opinion on the legalisation of same-sex marriage in Taiwan and elsewhere is an important and topical issue, and very much worthy of extensive social research, I did not feel the design of the study was well-thought out and justified to clearly contribute to new or important knowledge.
The focus on the difference between legalising same-sex marriage inside or outside the civil code seems quite a technical legal issue. This technicality seems to have been conflated too simplistically to survey respondants' personal perceptions around acceptability of same-sex marriage. The reason why the study is focused on this legal technicality is not well explained, nor how this technicality is of research significance.
Given that Taiwan eventually legalised same-sex marriage in May 2019 (in fact, under the civil code) after the referendum, the preferences for these legal technical differences among the public seems to dissapate as a significant issue. The fact that the paper did not emphasise the important fact that Taiwan has already legalised same-sex marriage until the conclusion seems confusing.
The focus on this legal technical issue also seems overall a little removed from the aim of this journal (i.e. public health, environment and global health). The references to psychological distress among non-heterosexual people in terms of public opinion on same-sex marriage seems important, yet not actually part of the research study, thus seems somewhat an unjustified claim.
The demarcation of survey participants into heterosexual versus non-heterosexual participants also seem to be an over-simplistic and over-presumptuous way of dividing the demographic make-up of the participants. Much more thorough analysis of other variables, such as age, gender, educaiton level, rural/urban, religion (which seems very important factor, as the authors suggest) would be needed to provide more through and rigorous data analysis. Some of this was briefly reported in the discussion section (4.5) but this was not well detailed and should be extrapolated in the results section instead, and given depth of analysis.
The collection of the survey in 'Wave 1' and 'Wave 2' form was not well justified. The fact that they seem to be different populations and different sample sizes make the two waves not comparable as longitudinal data.
The fact that Taiwan is the first Asian country to legalise same-sex marraige is no mean feat, and indeed this political and social acheivement should render much more exploration. Social and demographic context of Taiwanese culture, including political and historical trends and changes over time that may have influenced public opinion on same sex marriage seems a very important context to this topic. This paper unfortunately have fallen short on this. Much more emphasis on these issues in the introduction, study design and analysis would have made the paper much stronger.
While I commend the authors' research efforts on this important topic, due to the flaws in the design of the study, I regretfully recommend rejecting this manuscript.
Author Response
We appreciate your comments on our manuscript. As discussed below, we have revised our manuscript with underlines according to the reviewers. The following responses have been prepared to address your comments in a point-by-point fashion. Please let us know if there is anything else we should provide.
Comment
While public opinion on the legalisation of same-sex marriage in Taiwan and elsewhere is an important and topical issue, and very much worthy of extensive social research, I did not feel the design of the study was well-thought out and justified to clearly contribute to new or important knowledge.
Response
- Thank you for your comment. This study recruited participants from the Facebook advertisement. Access to Facebook is not yet universal, and all people are not equally motivated to use Facebook. We listed it as one of limitations of this study. Please refer to line 4-5. We also added “A Facebook Survey Study” into the topic to remind the readers the method of recruiting participants. Please refer to line 421-424.
- In spite of the limitation mentioned above, the results of the present study answered the three research questions we proposed, including comparing the rates of preference of laws legalizing same-sex relationships between people with various attitudes toward legalization of same-sex relationships, comparing the rates of preference of laws legalizing same-sex relationships among people between the first and second waves of the survey, and examining the factors related to the preference of laws legalizing same-sex relationships among heterosexual and nonheterosexual people. To the best of our knowledge, the present study is the first one to examine these issues. We believe that the results can provide experiences for other countries that may hold referendums to determine what forms of laws should be established for the legal recognition of same-sex relationships in future.
Comment
The focus on the difference between legalising same-sex marriage inside or outside the civil code seems quite a technical legal issue. This technicality seems to have been conflated too simplistically to survey respondants' personal perceptions around acceptability of same-sex marriage. The reason why the study is focused on this legal technicality is not well explained, nor how this technicality is of research significance.
Response
We considered it important to examine the issue of establishing same-sex couple laws outside the Civil Code or changing the Civil Code to include same-sex marriage laws based on two reasons.
- First, research has found that the legalization of same-sex marriage through direct initiatives or referendums bestows greater benefit than civil unions or domestic partnerships (Herek, 2006). We have listed it in “1.2. Forms of laws for the legal recognition of same-sex” Please refer to line 77-78.
- Second, it reflected one of core issues of legalizing same-sex relationships since the debate on the Marriage Equality Bill in the Legislative Yuan in October 2016, especially in the We added a paragraph to introduce the debates on the forms of laws to legalize same-sex relationship in Taiwan as below. Please refer to line 97-105.
“In addition to the debates on whether same-sex relationships should be legalized, what kinds of laws Taiwan should legislate for same-sex relationship are also the focus of debates in the public hearings hold by the Legislative Yuan and mass media. Two forms of laws have been commonly discussed. The supporters of establishing same-sex couple laws outside the Civil Code argued that it takes the rights of same-sex couples into consideration and keeps the rights of heterosexual couples intact, whereas the supporters of changing the Civil Code to include same-sex marriage laws argued that establishing same-sex couple laws outside the Civil Code without changing the Civil Code itself was virtually discrimination against same-sex couples, as were the separate buses for white and black people in operation across the South of the United States in the 1950s [12].”
Comment
Given that Taiwan eventually legalised same-sex marriage in May 2019 (in fact, under the civil code) after the referendum, the preferences for these legal technical differences among the public seems to dissapate as a significant issue. The fact that the paper did not emphasise the important fact that Taiwan has already legalised same-sex marriage until the conclusion seems confusing.
Response
- Taiwan eventually legalized same-sex relationship in May 2019 outside but not under the Civil Code. Please refer to: https://law.moj.gov.tw/ENG/LawClass/LawAll.aspx?pcode=B0000001. Moreover, Taiwan legalized same-sex relationships but not same-sex marriage because the results of the referendums forbad legalizing same-sex marriage.
- In May 2017, Taiwan’s Council of Grand Justices announced that the current Civil Code that barred same-sex marriage was a violation of human rights to equality and was unconstitutional and stipulated that same-sex marriage should be legalized within 2 years in Taiwan. Hence, legalizing same-sex relationship has been decided since May 2017. We explained it in “3. Battle for the legal recognition of same-sex relationships in Taiwan” and please refer to line 106-108.
- The present study was conducted before legalization of same-sex relationships in Taiwan. Examining this issue benefits accumulating knowledge to legal recognition of same-sex relationship. We added an paragraph in Introduction section to introduce why the present study was necessary. Please refer to line 126-133.
“Social debates on legal recognition of same-sex relationship in Taiwan lasted for at two years. For most of people in Taiwan, it is the first time to seriously consider legal recognition of same-sex relationship. It is a good opportunity to examine what factors might influence Taiwanese people’s preference of laws legalizing same-sex relationships. The results of the survey may provide an explanation for the people’s attitudes toward the legal recognition of same-sex relationships, as well as may provide knowledge and experiences for other countries that may hold referendums to determine what forms of laws should be established for the legal recognition of same-sex relationships in future.”
Comment
The focus on this legal technical issue also seems overall a little removed from the aim of this journal (i.e. public health, environment and global health). The references to psychological distress among non-heterosexual people in terms of public opinion on same-sex marriage seems important, yet not actually part of the research study, thus seems somewhat an unjustified claim.
Response
Public health has been defined as the science and art of preventing disease, prolonging life and improving quality of life through organized efforts and informed choices of society, organizations, public and private, communities and individuals. As we have introduced in “1.1. Same-sex marriage bans and legal recognition of same-sex relationships,” same-sex marriage bans were associated with increased rates of psychiatric disorders in sexual minority people, whereas legal recognition of same-sex relationships can reduce discrimination against LGB individuals and bestows substantial psychological, social, and health benefits to individuals from sexual minorities. Research has also found that the legalization of same-sex marriage through direct initiatives or referendums bestows greater benefit than civil unions or domestic partnerships (Herek, 2006). Therefore, this study filled the aims and scope of International Journal of Environmental Research and Public Health (https://www.mdpi.com/journal/ijerph/about) well.
Comment
The demarcation of survey participants into heterosexual versus non-heterosexual participants also seem to be an over-simplistic and over-presumptuous way of dividing the demographic make-up of the participants.
Response
The demarcation of participants into heterosexual versus non-heterosexual participants was based on our research aim. Please refer to line 155.
Comment
Much more thorough analysis of other variables, such as age, gender, educaiton level, rural/urban, religion (which seems very important factor, as the authors suggest) would be needed to provide more through and rigorous data analysis. Some of this was briefly reported in the discussion section (4.5) but this was not well detailed and should be extrapolated in the results section instead, and given depth of analysis.
Response
In the present study we examined the difference in the preference of laws among participants of various gender, age, personal belief toward the importance of the legal recognition of same-sex relationships to human right equality and the social status of sexual minorities, and perceived social attitudes toward legalization of same-sex relationships. We did not survey religion because that the main religion beliefs in Taiwan, Buddhism and folk belief systems have neutral attitudes toward homosexuality. Christians comprise approximately 2.5%–6.5% of the population in Taiwan (Chu, 2016; Ministry of the Interior, Taiwan, 2016). However, Christians who are against legalization of same-sex relationships do their best to influence Taiwanese people’s attitudes toward legalization of same-sex relationships. Therefore, we examined perceived social attitudes toward legalization of same-sex relationships of participants but not their religion beliefs. Moreover, research also found that improvement in education leads to an attitude of social tolerance toward homosexuality in Taiwan [Cheng, Y.A.; Wu, F.C.F.; Adamczyk, A. Changing attitudes toward homosexuality in Taiwan, 1995–2012. Chin. Social Rev. 2016, 48, 317–345.]. Therefore, we examined personal belief toward the importance of the legal recognition of same-sex relationships but not education levels. The two-year social debates on legalizing same-sex relationships in Taiwan proceeded in mass and social media. People received message and communicate attitudes through the Internet and beyond the boundary of geographic areas. Therefore, we considered personal and perceived social attitudes were more important than the effect of rural/urban.
Comment
The collection of the survey in 'Wave 1' and 'Wave 2' form was not well justified. The fact that they seem to be different populations and different sample sizes make the two waves not comparable as longitudinal data.
Response
- The term “wave” is not specific to longitudinal studies. For example, the World Values Survey have conducted seven waves of survey since 1981. Each wave of survey has different participants. Please refer to: http://www.worldvaluessurvey.org/WVSDocumentationWV6.jsp.
- Although the participants in the Wave 2 survey were less than those in the Wave 1 survey, the sample size in Wave 2 survey was large enough to do statistical analysis and provided answers to the study questions.
Comment
The fact that Taiwan is the first Asian country to legalise same-sex marraige is no mean feat, and indeed this political and social acheivement should render much more exploration. Social and demographic context of Taiwanese culture, including political and historical trends and changes over time that may have influenced public opinion on same sex marriage seems a very important context to this topic. This paper unfortunately have fallen short on this. Much more emphasis on these issues in the introduction, study design and analysis would have made the paper much stronger.
Response
- As we have mentioned above, Taiwan is the first Asian country to legalize same-sex relationships but not same-sex marriage. We considered it as an important social, health and human right issue but not “feat.”
- Some critics contributed legalization of same-sex relationships in Taiwan to political achievement of the ruling party, Democratic Progressive Party (DPP). In fact, the DPP government had an ambiguous attitude toward legalization of same-sex relationships. For example, though President Tsai once stated in her personal Facebook during her 2015 presidential campaign, “I am Tsai Ing-Wen, and I support marriage equality,” she has not expressed support as president after being elected. Before the referendum, the DPP government had to originally detail the administration’s stance prior to the referendum, as mandated in the Referendum Act, but the administration led by President Tsai and William Lai, President of the Executive Yuan, kept delaying their explanation of the referendum and absented themselves from referendum debates. DPP legislators’ support the Enforcement Act of Judicial Yuan Interpretation No. 748 primarily because of partisan reasons rather than because of their support for same-sex marriage. Hence, it is not reasonable to attribute legalization of same-sex relationship to the DPP government’s political achievement.
- It is also hard to attribute legalization of same-sex relationship to social achievement in Taiwan. The 2012 Taiwan Social Change Survey showed that for the first time, supporters of same-sex marriage outnumber those who oppose it [Chang, Y.H.; Tu, S.H.; Liao, P.S. Taiwan Social Change Survey 2012, Phase 6, Wave 3; Academia Sinica: Taiwan, Institute of Sociology, 2013.]. However, the results of vote indicated that the referendums against same-sex marriage received overwhelming support compared with the referendum supporting same-sex marriage.
- Based on the facts described above, the present study focused on the relationships between personal and perceived attitudes toward legalizing same-sex relationships but not political or social influences and the preference of laws in Taiwan.
Comment
While I commend the authors' research efforts on this important topic, due to the flaws in the design of the study, I regretfully recommend rejecting this manuscript.
Response
We have responded to your comments in a point-by-point fashion. We hoped that our explanations can provide you a basis to change your recommendation to the editors of the journal.
Round 2
Reviewer 3 Report
I'd like to thank the authors for updating the manuscript and addressing my comments. Some aspect of the manuscript have improved (e.g. contextualise the socio-cultural context in Taiwan) but the overall design of the study still needs better justification and clarity. Reading the new revision I felt that readers would find this a very confusing study to follow because of the various complex technicalities and poor justificaiton of the research topic and design.
1) Taiwan eventually legalised same-sex marriage in May 2019 - this is widely reported in Western media, and this is what most people outside of Taiwan understand, and would expect some contextualisation and explanation as to how this study is relevant to this. (a) This important fact should be discussed up front (e.g. introduction), as it seems misleading without outlining this. (b) This survey predates this law passage. How this survey may help clarify how the eventual law passage needs to be explained. (c) If legalisation of marriage is not in the civil code, as the authors claimed, then this seems really important point to discuss - e.g. in both the introduction and the conclusion.
2) Reporting the the age breakdown of the survey resopndants into three brackets: "20 - 29; 30 - 39; 40 and older" This seems a very crude and unsophisticated bracket: 40+ is a huge population, and most studies would break down at least up to the age of 80s. e.g. "40 - 49; 50 - 59; 60 - 69, 70 - 79; 80 and above" etc.
3) There is still no clear justification why "heterosexual" and "nonheterosexual" people are compared (Aim 3). Why is this an important comparason? What does this aim achieve? How is this supported by any previous literature? How does this add to overall understanding of the "big picture" issue of discrimination, human rights etc.
4) While the study results does seem to meet its aim by default, the point is, the study is not well designed or justified, and thus the conclusions does not serve the purpose of adding to good quality new knowledge.
Overall I found the aims and justification of the study quite difficult to read and follow. The focus on legal technicalies in this survey and how this fits in with the big picture discussions around discrimination, human rights, cultural attitudes etc needs much better linking. My impression is that the survey actually over-complicates the big picture of social attitudes with a highly technical survey design that does not serve the 'big picture' purpose of what the study seems to want to achieve. There are too many different tangents in the narrative of the paper to help the reader clearly understand the key point. "Keep it simple" is my main suggestion.
Author Response
Reviewer 3
We appreciate your comments on our manuscript. As discussed below, we have revised our manuscript with underlines according to the reviewers. The following responses have been prepared to address your comments in a point-by-point fashion. Please let us know if there is anything else we should provide.
Comment
1) Taiwan eventually legalised same-sex marriage in May 2019 - this is widely reported in Western media, and this is what most people outside of Taiwan understand, and would expect some contextualisation and explanation as to how this study is relevant to this. (a) This important fact should be discussed up front (e.g. introduction), as it seems misleading without outlining this. (b) This survey predates this law passage. How this survey may help clarify how the eventual law passage needs to be explained. (c) If legalisation of marriage is not in the civil code, as the authors claimed, then this seems really important point to discuss - e.g. in both the introduction and the conclusion.
Response
- a) In Introduction, we added a paragraph as below to introduce how the Taiwanese people’s preference of laws for the legal recognition of same-sex relationships shown in the referendums influenced the legislation of the Enforcement Act of Judicial Yuan Interpretation No. 748 enacted by the Taiwanese government for legalizing same-sex relationship. Please refer to line 125-129.
“Finally, the Taiwanese government enacted the Act for Implementation of Judicial Yuan Interpretation No. 748 outside the Civil Code in May 2019. This law was the effort of Taiwanese government to seek a compromise between the Constitutional Court’s interpretation and the referendum results by guaranteeing most of the same rights entailed in a heterosexual marriage for same-sex couples.”
- b) We also added the sentences as below explaining that this survey may help clarify how the eventual law passage. Please refer to line 130-132.
“Taiwanese people’s preferences of laws shown in the referendums deeply influenced the final result of legislation. It is important to survey what factors related to the preferences of laws and what changes of the preferences happened during the social debates on legalizing same-sex relationship.”
- c) We mentioned that legalization of same-sex relationship is not in the civil code in Introduction as described in a) (please refer to line 126) and conclusion (please refer to line 444).
Comment
2) Reporting the the age breakdown of the survey resopndants into three brackets: "20 - 29; 30 - 39; 40 and older" This seems a very crude and unsophisticated bracket: 40+ is a huge population, and most studies would break down at least up to the age of 80s. e.g. "40 - 49; 50 - 59; 60 - 69, 70 - 79; 80 and above" etc.
Response
This study recruited participants via Facebook. Like other studies recruiting participants via Facebook, the participants in this study tended to be young. Only 12.9% and 19.6% of the participants were 40 or older in the Wave 1 and Wave 2 surveys, respectively. If we broke down the participants according age up to the age of 80s, the numbers of participants would be too small. In the revised manuscript we listed the overrepresentation of young adults as one of the limitations of this study as below. Please refer to line 437-441.
“Moreover, like other studies recruiting participants via Facebook [25,26], the participants in this study tended to be young, and only 12.9% and 19.6% of the participants were 40 or older in the Wave 1 and Wave 2 surveys, respectively. Age has been identified as a factor significantly related to the level of tolerance for homosexuality in Taiwan [12].”
Comment
3) There is still no clear justification why "heterosexual" and "nonheterosexual" people are compared (Aim 3). Why is this an important comparason? What does this aim achieve? How is this supported by any previous literature? How does this add to overall understanding of the "big picture" issue of discrimination, human rights etc.
Response
The present study did not compare the preference of law legalizing same-sex relationships between heterosexual and nonheterosexual people. Because that legal recognition of same-sex relationship is directly related to the rights of nonheterosexual people, it is apparent that the aspects and attitudes toward this issue in nonheterosexual people will be different from those in heterosexual people. Therefore, the present study examined the factors related to the preference of laws in heterosexual and nonheterosexual people separately. We explained it in Introduction section, line 159-163.
“Because that legal recognition of same-sex relationship is directly related to the rights of nonheterosexual people, it is apparent that the aspects and attitudes toward this issue in nonheterosexual people will be different from those in heterosexual people.; therefore, the present study examined the factors related to the preference of laws in heterosexual and nonheterosexual people separately.”
Comment
4) While the study results does seem to meet its aim by default, the point is, the study is not well designed or justified, and thus the conclusions does not serve the purpose of adding to good quality new knowledge.
Response
As we mentioned in the last response to the comment, the present study is the first one to examine the rates of preference of laws legalizing same-sex relationships between people with various attitudes toward legalization of same-sex relationships, to compare the rates of preference of laws among people before and after the referendums, and to examine the factors related to the preference of laws legalizing same-sex relationships. We believe that the results can provide experiences in Taiwan for other countries that may hold referendums to determine what forms of laws should be established for the legal recognition of same-sex relationships in future. We hope that people who are interested to the issue have the chance to see our study and make comments to it.
Comment
Overall I found the aims and justification of the study quite difficult to read and follow. The focus on legal technicalies in this survey and how this fits in with the big picture discussions around discrimination, human rights, cultural attitudes etc needs much better linking. My impression is that the survey actually over-complicates the big picture of social attitudes with a highly technical survey design that does not serve the 'big picture' purpose of what the study seems to want to achieve. There are too many different tangents in the narrative of the paper to help the reader clearly understand the key point. "Keep it simple" is my main suggestion.
Response
- Legal recognition of same-sex relationships was never been a simple issue. Various countries have various sociocultural attitudes toward legalization of same-sex relationship. We think that it is necessary to introduce what debates have happened regarding to legalizing same-sex relationship in Taiwan to assist the readers in understanding.
- The present study focused on three study aims, and these three aims were coherent. The contents of the results and discussion correspond to the three study aims. We also used subtitles to assist the readers in master the points.